# An Exploratory Bioinformatic Investigation of Cats’ Susceptibility to Coronavirus-Deriving Epitopes

**DOI:** 10.3390/life14030334

**Published:** 2024-03-02

**Authors:** Michela Buonocore, Davide De Biase, Domenico Sorrentino, Antonio Giordano, Orlando Paciello, Anna Maria D’Ursi

**Affiliations:** 1Department of Veterinary Medicine and Animal Production, University of Naples Federico II, Via Federico Delpino, 1, 80137 Napoli, Italy; michela.buonocore@unina.it; 2Department of Chemical Sciences, Research Centre on Bioactive Peptides (CIRPeB), University of Naples Federico II, Complesso Universitario di Monte Sant′Angelo, Via Cintia, 80126 Naples, Italy; 3Department of Pharmacy, University of Salerno, Via Giovanni Paolo II, 132, 84084 Fisciano, Italy; ddebiase@unisa.it (D.D.B.); d.sorrentino14@studenti.unisa.it (D.S.); 4Sbarro Institute for Cancer Research and Molecular Medicine, Center for Biotechnology, College of Science and Technology, Temple University, Philadelphia, PA 19122, USA; president@shro.org; 5Department of Medical Biotechnologies, University of Siena, 53100 Siena, Italy

**Keywords:** coronavirus, MHC, peptides, immunity system

## Abstract

Coronaviruses are highly transmissible and pathogenic viruses for humans and animals. The vast quantity of information collected about SARS-CoV-2 during the pandemic helped to unveil details of the mechanisms behind the infection, which are still largely elusive. Recent research demonstrated that different class I/II human leukocyte antigen (HLA) alleles might define an individual susceptibility to SARS-CoV-2 spreading, contributing to the differences in the distribution of the infection through different populations; additional studies suggested that the homolog of the HLA in cats, the feline leukocyte antigen (FLA), plays a pivotal role in the transmission of viruses. With these premises, this study aimed to exploit a bioinformatic approach for the prediction of the transmissibility potential of two distinct feline coronaviruses (FCoVs) in domestic cats (feline enteric coronavirus (FeCV) and feline infectious peritonitis virus (FIPV)) using SARS-CoV-2 as the reference model. We performed an epitope mapping of nonapeptides deriving from SARS-CoV-2, FeCV, and FIPV glycoproteins and predicted their affinities for different alleles included in the three main loci in class I FLAs (E, H, and K). The predicted complexes with the most promising affinities were then subjected to molecular docking and molecular dynamics simulations to provide insights into the stability and binding energies in the cleft. Results showed the FLA proteins encoded by alleles in the FLA-I H (H*00501 and H*00401) and E (E*01001 and E*00701) loci are largely responsive to several epitopes deriving from replicase and spike proteins of the analyzed coronaviruses. The analysis of the most affine epitope sequences resulting from the prediction can stimulate the development of anti-FCoV immunomodulatory strategies based on peptide drugs.

## 1. Introduction

Coronaviruses (CoVs) are RNA viruses with a large genome and critical infectivity. The CoV family includes various viruses, with tropism for humans and animals. Among them, two distinct but very similar feline coronavirus (FCoV) serotypes are widespread in domestic cats: type 1 FCoV, or feline enteric coronavirus (FeCV), and type 2 FCoV, or mutated feline enteric coronavirus, also known as feline infectious peritonitis virus (FIPV) [1]. To date, these viruses are still considered fatal for young cats, and therapies are mainly based on antiviral nucleotides like remdesivir and its analogs. However, no prophylaxis or vaccine is available [2,3]; in fact, cases of antibody-dependent enhancement (ADE) of FIPV infectivity have been observed following the development of the humoral response, which facilitates the entry of the virus in the macrophages, where the pathogen amplifies its replication [4,5,6].

FeCV and FIPV share very similar genetic frameworks with the well-known SARS-CoV-2, a highly pathogenic virus for humans whose infection has also been reported in domestic animals but is relatively uncommon in cats [7,8]. Currently, the mechanisms underlying the susceptibility of certain animals to SARS-CoV-2 infection are still unclear. We previously demonstrated that the similarity with angiotensin-converting 2 (ACE2), the enzyme responsible for the virus entry into the host cells, might be a determinant [9].

The efficiency of the immune response to the viral infection, which is the basis of the virus pathogenicity, includes an interaction of viral antigens with the major histocompatibility complex (MHC). Two classes of MHC—classes I and II—operate in the capture of antigens deriving from pathogen proteins: class I MHCs bind 8–10 amino acids long antigen peptides to present them to CD8^+^ CD4^−^ cytotoxic T lymphocytes to trigger the cell-mediated immune response, while class II MHCs bind 12–24 amino acids long antigen peptides and introduce them to CD4^+^ CD8^−^ T helper cells to stimulate the humoral antibody production by B cells [10]. Pharmacologically, the importance of MHC as a target for immunodrugs has been widely assessed [11], and recently, many works reported the use of partial MHC constructs as modulators of T cells and CD74 signaling and, thus, neuroprotective agents in stroke [12,13,14,15,16].

In humans, the group of polymorphic genes encoding for MHCs is also referred to as human leukocyte antigens (HLAs). These cell surface receptors are characterized by high sequence variability, mainly in the antigen-binding region. The effectiveness of the immune response depends on the interactions of antigens in the binding pocket of the MHC and how these antigens are presented for interaction with T cells. Recently, it has been demonstrated that an individual susceptibility to SARS-CoV-2 is correlated with a specific combination of class I/II HLAs encoded by a specific allele combination. Notably, HLA-C*01 and B*44 alleles have been identified as potential genetic risk factors for COVID-19, contributing to the differences in the distribution of SARS-CoV-2 infection through different populations (e.g., Northern and Southern Italy) [17].

Analogous to human HLA, the feline leukocyte antigen (FLA) mediates the feline immune response by interacting with antigens derived from pathogens infecting cats [18]. However, it has been experimentally demonstrated that there are remarkable differences between HLA and FLA genomic structures: differently from HLA, whose classes are all located on the p arm of the human chromosome 6, FLA alleles are found mostly in the pericentromeric region on the q arm of the chromosome B2 in cats, yet the class I region is split and placed also in the peritelomeric region of the p arm of the same chromosome (Figure 1A) [19,20]. Despite these differences in the genomic organization, the sequences of the proteins encoded by class I FLA and HLA alleles share an identity of about 70–75%, and their 3D structures overlap with a low RMSD value (Figure 1B).

Considering the similarity between these viruses and SARS-CoV-2, in this work we aim to investigate with a bioinformatic approach whether a specific combination of FLA alleles can favor the individual susceptibility of some cats over others regarding exposition to FIPV and FeCV. Using SARS-CoV-2 as a model, for which a great amount of structural data are available, we aim to (i) identify alleles that potentially correlate with enhanced cell-mediated immunogenic response in domestic cats infected with FCoVs, (ii) identify epitopes in CoV proteins that are most likely to target the proteins encoded by these alleles, and (iii) conduct a peptide search for the epitopes on viruses infecting with tropism for different species to hypothesize any cross-reaction.

For this aim, we retrieved all the viral glycoprotein sequences belonging to SARS-CoV-2, FeCV, and FIPV to find epitope regions potentially recognized by the MHC receptors encoded in the E, H, and K loci of class I FLA (FLA-I). We selected these loci since they were found to restrict antiviral CD8^+^ T-cell effectors [21], whose response has already been observed to control FIV infections [22,23,24]. Using this biocomputational approach, whose workflow is described in Figure 2, it was possible to identify six interesting epitopes on viral glycoproteins, together with the most affine proteins encoded by FLA-I alleles that can define a subjective susceptivity of cats with a peculiar genetic makeup over others.

## 2. Materials and Methods

### 2.1. Epitope Mapping, Sequence Alignment, and Peptide Search

The epitope mapping was performed using the online server NetMHCPan 4.1 [25], which predicts the affinity of viral epitopes for MHCs using artificial neural networks. The viral glycoprotein sequences retrieved from the UniProt database [26] were uploaded as FASTA type. We selected to restrain nonapeptides from the proteins according to the literature, which suggests that 9-mers are more affine for class-I FLAs [18,21,27]. The sequences of FLA-I E, H, and K loci were manually retrieved from UniProt and uploaded to NetMHCPan. The results were analyzed according to the mass spectrometry elution score (EL), selecting the epitopes with scores above 0.5.

The sequences with an EL score above 0.8 were aligned using the *Multiple Sequence Alignment* tool included in Maestro 2023-2 [28]. The search for the epitope sequences was performed using BLAST [29] included in UniProt, and the peptides with an identity of 90–100% were considered.

### 2.2. Molecular Docking

Molecular docking calculations were carried out with HPepDock [30] for the ab initio building of the binding complexes and with AutoDock CrankPep (ADCP [31]) for the refinement. FLA-I 3D structures retrieved from UniProt were set as receptor proteins. The grid boxes were centered on the residues included in the two helices that overhung the large binding site. The HPepDock results were used as reference for the refinement of the binding poses performed by ADCP. The peptides were sampled in extended and helix conformations. Results were analyzed according to the lowest docking scores (kcal/mol), which indicate the binding poses with the most favorable interaction energies, and the lowest values of RMSD, which represent the reproducibility of the poses. Molecular visualization was performed with Maestro [28].

### 2.3. Protein Structure Prediction with AlphaFold2

The 3D structure prediction of the unknown moieties of R1ab protein was performed with AlphaFold2 [32] included in the ColabFold server [33] with the parameters’ multisequence alignment mode (MSA mode) set as mmseqs2_uniref_env and pair_mode set as unpaired_paired. The models were ranked according to the predicted local distance difference test (pLDDT) score and the predicted alignment error (PAE), indicating a distance score between pairs of residues with low values specifying low errors.

### 2.4. Molecular Dynamics

To perform MD simulations, we prepared the ADCP deriving complexes using the *Protein Preparation Wizard* tool included in Maestro [28], which allowed for adding the side chains in the peptides, minimizing the energies, and optimizing the bond angles and lengths. MD simulations were run using GROMACS 2020.3 [34]. The topology files were generated using the CHARMM36 all-atom force field [35]. The complexes were solvated in cubic boxes with the TIP4P water model. Na^+^ and Cl^−^ ions were added to neutralize the charge of the system. After minimization using the steepest descent integrator, the system was equilibrated at the average body temperature in cats of 311.65 K for 1 ns as NVT ensemble and at 1 atm pressure using Berendsen algorithm NpT ensemble for 1 ns. The outputs were used for an MD simulation using particle mesh Ewald for long-range electrostatics under NpT conditions. Coordinates were saved every 100 ps. Trajectory files containing the coordinates of the receptor–ligand complex at different time steps (from 100 ps to 10 ns) were fitted in the box and converted in PDB coordinates by using the *trjconv* tool of GROMACS. The structures were visualized with Maestro by Schrödinger [28]. Analyses of RMSD, number of bonds (H-bonds and neighbors within 0.35 nm), and short-range interaction energies (Coulomb and Lennard-Jones) between the two energy groups (set as receptor and peptides) were carried out for the MD simulations of each system using the *rms*, *hbond*, and *energy* tools of GROMACS.

## 3. Results

### 3.1. Epitope Mapping

We performed an exploratory epitope mapping using the online server NetMHCPan 4.1 [25]. To predict specific bindings of MHC with epitopes of any length, the service requires the amino acid sequences of the FLA-I of interest. For this purpose, among the entries in the UniProt database [26] marked as “reviewed”, i.e., entries with manually annotated records, we selected 12 variants for FLA-I E, 11 for FLA-I H, and 9 for FLA-I K (Appendix A). Concerning the restrained peptide sequences, we considered only the entries marked as “reviewed” of the viral glycoproteins from the three CoVs under scrutiny: SARS-CoV-2, FIPV, and FeCV (Appendix A). The epitope mapping was carried out by restraining the search to nine amino acid long epitope peptides [18,21,27]. Outcomes were filtered according to their mass spectrometry eluted ligand (EL) score. Two EL score cut-offs were considered to select the best predictions: 0.5 for all the best results and 0.8 for a refinement of the outputs.

We obtained outputs as follows:

(a) The data relative to the 12 FLA-I E allele variants indicate that the protein encoded by the E*00101 allele binds viral epitopes with the best EL scores: 41 having an EL score > 0.8 and 332 > 0.5. For the remaining alleles in the E locus, very few epitopes are selected with a score > 0.8 (average 3), while a significant number of epitopes are selected with an EL score > 0.5 (average 113) (Appendix A).

(b) Sequences encoded by FLA-I H variants generally interact with a significant number of epitopes with scores > 0.8 (average 25) and a high number with scores > 0.5 (average 334). Epitopes characterized by the best EL scores bind MHC proteins encoded by H*00401, H*008012, H*00701, and H*00501 alleles (Appendix A).

(c) The data relative to FLA-I K variants indicate that six out of the nine variants under scrutiny bind none of the epitopes with scores > 0.8 and few epitopes with scores > 0.5 (average 44). Interestingly, the FLA-I K*00801 allele encodes for a sequence binding the highest number of epitopes with a score > 0.9 (15) and, in general, the highest number of peptides with a score > 0.5 (310) (Appendix A).

Regarding the search of epitopes from viral glycoproteins, the analysis indicated that the peptides with the highest EL scores were derived from the spike (S) protein (P0DTC2-SARS CoV-2) (P10033-FIPV) and the replicase 1ab (R1ab) protein (P0DTD1/P0DTC1-SARS CoV-2) (Q98VG9-FIPV). Research on FeCV proteins did not produce epitopes with significant EL scores.

### 3.2. Sequence Alignment

In search of consensus sites, including amino acids essential for interacting with FLA-I receptors, we performed a sequence alignment of the epitopes found, considering the amino acid sequences with the highest EL scores as templates. Figure 3 shows the frequency of finding a given residue in each position for the epitopes that resulted in binding the three loci of FLA-I with an EL score > 0.8. Accordingly, proline shows a frequency > 80% to occupy position 2 for epitopes recognized by proteins encoded in the E and K loci. An aromatic residue is highly recurrent in position 9 for epitopes bound by proteins encoded in the E and H loci. Among the epitopes recognized from proteins encoded in the H locus, we identified a set of sequences exhibiting common motifs, consisting of phenylalanine in 3, aspartate in 4, and apolar residues in positions 5 and 6. All the epitopes of the cluster include an aromatic residue at position 9. However, we can identify two subsets, one characterized by an apolar residue in 8 and tryptophan in 9, the other characterized by lysine in 8, and phenylalanine or tyrosine in 9.

This alignment allowed us to select six patterns of sequences having amino acids with a frequency > 50%: X-P-X(6)-V, X-P-X(6)-L, X-P-X(6)-Y, X-S-X(6)-Y, X-S-X(6)-F, X-A-X(6)-W. We performed a search for these patterns using ScanProsite included in the server Prosite [36] on a randomly generated list of 2000 glycoproteins of viruses targeting *Felis catus* available in the NCBI Virus database [37]. The results report that all the templates are frequently retrieved in the glycoproteins, in particular the feline immunodeficiency virus (FIV) gp100, env, and pol proteins; feline calicivirus (FCV) capsid protein; felis domesticus papillomavirus (FdPV) L2 protein; feline adenovirus (FeAdV) hexon protein; and feline picornavirus (FePV) polyprotein.

### 3.3. Molecular Docking

Peptide sequences having EL scores > 0.8 were subjected to molecular docking calculations against the FLA-I proteins. The docking simulation of peptides is often challenging using the conventional programs for docking small molecules because of the high number of rotatable bonds that are not correctly sampled. To minimize this issue, we used two peptide-optimized software, HPepDock [30] for the ab initio generation of the complexes and AutoDock CrankPep (ADCP [31]) for the refinement of the poses. The peptides were sampled in helix and extended conformations. Table 1 reports all the binding poses of the epitopes resulting in a docking score lower than -10 kcal/mol in their extended conformation. Among the epitopes, the sequence _3574_RTIKGTHHW_3582_ deriving from SARS-CoV-2 R1a exhibits the best binding parameters in both conformations against the proteins encoded by the alleles FLA-I H*00501, H*00401, H*008012, and FLA-I K*00701. Additionally, the epitope _6437_KQFDTYNLW_6445_, derived from SARS-CoV-2 R1ab, is characterized by a consistent binding with FLA-I K*00701 and FLA-I H*00501. Likewise, the epitopes _3756_AANELNITW_3764_ and _4533_RLYYETLSY_4541_, deriving from the replicases of FIPV, interact with remarkable affinity with FLA-I H*00501. Overall, most of the peptides derived from R1ab are characterized by favorable binding parameters against alleles derived from the FLA-I H and K loci. On the other hand, the sequences deriving from the spike proteins showed milder yet favorable binding parameters, especially with alleles of the FLA-I E and H loci: considering a docking score lower than -15.0 kcal/mol, among FIPV S-deriving epitopes, _1325_RPNWTVPEF_1333_ was predicted to have good interactions with FLA-I E*00701 and E*00101, _771_TTTPNFYYY_779_ with H*00501, and _1228_TAYETVTAW_1236_ with H*00401. Regarding SARS-CoV-2 S-deriving epitopes, _625_HADQLTPTW_633_ reported the best docking scores in complex with the proteins encoded by the two alleles E*01001 and E*00101, as well as H*008012 and H*00401. Conversely, _321_QPTESIVRF_329_ reported the best results in complex with the protein encoded by E*00701.

Following these results, the above-mentioned epitopes with the best docking score results will be mentioned in the text as reported in Table 2:

### 3.4. Analysis of Possible Surface Exposition

Analyzing the exposition of the epitopes on a protein is crucial to evaluate their potential immunogenicity and eligibility to be recognized by immunoglobulins. Unfortunately, very little is known about the structural information of the R1ab moieties containing the sequences potentially identified as epitopes. However, it is possible to use SARS-CoV R1ab as a template to analyze the location of the epitopes on the protein surface. Accordingly, the structure having PDB ID: 6NUS [38] was used to locate _fipv-r_Ep4.

Figure 4 shows the sequence alignment of the R1ab of SARS-CoV and FIPV and the possible localization of the _fipv-r_Ep4 sequence in the folded protein.

Unfortunately, no experimental structural data matched _sars-r_Ep1, _sars-r_Ep2, and _fipv-r_Ep3. Therefore, we predicted these moieties using AlphaFold2 [32,33] and located the epitope on the best-ranked models (Appendix A), which report a rather good exposition of the epitopes (Figure 5).

More experimental structural information is available on the SARS-CoV-2 S protein, but no structural information is available on FIPV S. Therefore, we aligned FIPV and SARS-CoV-2 S sequences to locate the three FIPV S-deriving epitopes. Then, we searched for the corresponding sequences on the reference SARS-CoV-2 S 3D structure (PDB ID: 7WEB [39]). Figure 6 shows the possible localization of the epitopes. Interestingly, _fipv-s_Ep8 is partially aligned with the SARS-CoV-2 S-deriving epitope _sars-s_Ep5 (Figure 6B). Additionally, the epitope _fipv-s_Ep9 is well exposed on the protein surface, while _fipv-s_Ep7 is partly embedded in the transmembrane domain.

### 3.5. Molecular Dynamics

The stability of the MHC/peptide complex is essential to develop the immunogenic response. To evaluate the stability of the binding over time, the FLA proteins in complex with some of the peptides previously identified were subjected to 50 ns classical MD simulations in water at the average body temperature of cats (311.65 K). We focused on the epitopes that in the previous analyses reported the best docking scores and the most convenient expositions on the viral glycoprotein surfaces. Accordingly, the binding complexes selected were as follows:_sars-r_Ep1 with FLA-I H*00501 for the outstanding docking score (–22.6 extended (e) and –22.5 helix (h));_fipv-r_Ep4 with FLA-I H*00501 for the docking score (–19.8 e and –20.2 h) and the possible localization on the viral glycoprotein surface;_sars-s_Ep5 with FLA-I E*01001 and _sars-s_Ep6 with FLA-I E*00701 for the docking scores (–16.3 e and –14.5 h; –15.6 e and –16.3 h) and the position on the viral glycoprotein surface;_fipv-s_Ep8 with FLA-I H*00501 and _fipv-s_Ep9 with H*00401 for the docking scores (–16.5 e and –16.3 h; –15.5 e and –15.5 h) and the possible localization on the viral glycoprotein surface.

A total of 12 simulations were run using the peptides in extended and helix conformations derived from the lowest energy docking poses.

Based on the results in Appendix A, all the epitopes (except for _sars-s_Ep6) steadily interact with the receptors throughout the simulations in extended samplings, while in helix conformation, the RMSD values do not always indicate a well-equilibrated system (Appendix A). Moreover, only _sars-r_Ep1 and _fipv-s_Ep8 bind it in a helix conformation. Overall, the peptides sampled with extended conformations result in higher numbers of established H-bonds and neighbor contacts. Despite the relatively short sequences, peptides bind with low energies and occupy a wider surface in the large MHC binding site.

### 3.6. Peptide Search

We performed a peptide search of the epitopes _sars-r_Ep1, _fipv-r_Ep4, _sars-s_Ep5, _sars-s_Ep6, _fipv-s_Ep8, and _fipv-s_Ep9 in UniProt to understand if the epitopes are unique or repeated on other viral glycoproteins. Our inquiry showed that the SARS-CoV-2-deriving epitopes _sars-r_Ep1, _sars-s_Ep5, and _sars-s_Ep6 are unique and only retrieved with little differences in the sequences of other CoVs like bat and pangolin ones. On the other hand, the amino acid composition of the FIPV-deriving epitopes is present in different viruses. In particular, _fipv-r_Ep4 and _fipv-s_Ep9 are present in canine, porcine, and mink CoVs, while _fipv-s_Ep8 is present in canine and porcine CoVs (Table 3).

## 4. Discussion and Conclusions

The high transmissibility and pathogenicity of SARS-CoV-2 in humans have been two of the main factors causing the onset of the COVID-19 pandemic, yet SARS-CoV-2 infection has also been reported in domestic animals [7,8]; however, human-to-animal transmission seems to be an infrequent event [7,9]. Recently, it has been demonstrated in humans that different class I/II HLA alleles may define a distinct susceptibility to SARS-CoV-2 and its spreading among different populations [17]. These data suggest that, considering the high homology between SARS-CoV-2 and FCoVs genomes, the feline homolog of the HLA, the FLA, may play a role in the transmission of CoVs more commonly reported in this species [1]. However, while for SARS-CoV-2 monoclonal antibodies appear to be effective in controlling the spreading of the COVID-19 disease [40], in FCoVs, an important adverse reaction has been observed when trying to develop the humoral response, namely, ADE, which causes a critical improvement of the viral replication in the macrophages [4,5,6].

Following a similar approach, in this work, we presented a bioinformatic investigation to understand the individual susceptibility of domestic cats to develop a cell-mediated immune response after being exposed to epitopes deriving from FeCV and FIPV glycoproteins. Given the large availability of structural information, analysis was performed using SARS-CoV-2 proteins as reference models. This study also aimed to understand the structural keys regulating the interaction of these epitopes with the proteins encoded by the different feline MHC alleles and inducing a distinctive cell-mediated response. The results may allow the design of a strategy to avoid the drawback of ADE derived from the humoral response.

As a first step, an epitope mapping of nonapeptides deriving from viral glycoproteins targeting several proteins encoded by alleles of the FLA-I E, H, and K loci allowed for the selection of the most affine sequences and the most suitable alleles; in fact, this initial analysis indicated that the best resulting epitopes derive from R1ab and S glycoproteins of SARS-CoV-2 and FIPV and that FLA-I H alleles express the most responsive receptors. Analysis of peptide sequences by sequence alignment indicated that in the epitopes with the highest affinity for the protein expressed from FLA-I E and H alleles, it is rather frequent to find an aromatic residue at the end of the sequence, while a proline in position 2 is quite common in epitopes targeting FLA-I E and K. This finding might help in the identification of other viral glycoproteins that are potentially immunogenic in cats, by searching for sequence motifs where proline and aromatic residues are appropriately spaced. Molecular docking of all the epitopes having an EL score > 0.8 with the corresponding receptor evidenced good binding scores for epitopes deriving from the R1ab of SARS-CoV-2 and FIPV in complex with FLA-I H alleles (in particular FLA-I H*00501) and less potent but still favorable binding scores for epitopes deriving from the R1ab of SARS-CoV-2 and FIPV S-deriving peptides with FLA-I E and H alleles. To further filter these results, we examined the possible exposition of the epitopes on the protein surface since a better exposition is needed for an epitope to be recognized by the immune system. This analysis led to the selection of well-exposed epitopes on the R1ab surface and, at the same time, to the exclusion of epitopes that are almost embedded in the transmembrane domain of the S protein; moreover, it permitted the discovery that _sars-s_Ep5 and _fipv-s_Ep8 are possibly situated in the same region, suggesting a high immunogenic potential of this portion. Accordingly, the two mentioned epitopes are located upstream of the receptor-binding domain, which is already a known immunogenic moiety for humans [41,42,43]. A total of six epitopes were submitted to 50 ns MD simulations in complex with the most affine FLA as derived from docking. The epitopes were sampled in the extended and helix conformations to evaluate the effect of the secondary structure on the binding. The results showed that most peptides prefer an extended conformation for a favorable binding: the interaction energies (Coulomb and Lennard-Jones) describe overall stable complexes with a fewer number of spikes in the energy plots due to a higher number of established bonds (Appendix A) when the peptides are sampled in the extended conformations, thus managing better to occupy the extended binding cleft of the receptor. Eventually, the study suggests that the FLA-H locus is mostly affine for R1ab and FIPV S-deriving epitopes, while the FLA-I E locus has good affinity for SARS-CoV-2 S peptides. To understand the possible implications of the epitopes in heterogeneous serological relationships among different coronaviruses, we searched for the six epitope sequences in the UniProt database. SARS-CoV-2-deriving epitopes are rather unique, and slightly different sequences are found in glycoproteins of bat (RaTG13) and pangolin (PCoV_GX) coronaviruses, which are closely related and involved in the evolution and cross-species transmission of the virus [44]. However, these sequences are strictly conserved among all the variants of interest and concern of SARS-CoV-2. FIPV-deriving epitopes were instead retrieved in canine enteric coronavirus (CCoV) and porcine transmissible gastroenteritis coronavirus (TGEV), confirming the widely demonstrated close relationship between these three coronaviruses [45]. To validate the six predicted sequences with experimental data, we performed a search for the obtained epitopes on the immune epitope database IEDB (www.iedb.org, accessed on 22 February 2024) [46], which reported that (i) _sars-r_Ep1 (IEDB ID: 2249103) has been identified by an AI prediction in a pool of sequences as an epitope for several class I HLA alleles and tested in a mix of peptides that acted as a CD8^+^ T-cell activator [47], (ii) _sars-s_Ep5 (IEDB ID: 1332221) has been found to bind in vitro different alleles of the HLA-I B locus [41], and (iii) _sars-s_Ep6 (IEDB ID: 1323461) has been extensively studied among HLA-I-binding restricted peptides for its ability to activate CD8^+^ T cells [48,49,50,51,52,53]. On the other hand, FIPV-derived _fipv-r_Ep4 and _fipv-s_Ep8 epitopes have not been found in the database, while the _fipv-s_Ep9 sequence is contained in a longer epitope (IEDB ID: 142156), which has been found to significantly enhance the feline interferon (IFN)-γ levels in peripheral blood mononuclear cells (PBMC) and has been identified as an antibody-binding epitope [54].

In conclusion, this investigation suggests that domestic cats expressing FLA-I H*00501 and H*00401 alleles might develop immune responses following exposition to epitopes deriving from the R1ab of FIPV and the S of FIPV. In contrast, those expressing FLA-I E*01001 and E*00701 might be more sensitive to epitopes similar to those deriving from SARS-CoV-2 S. Though SARS-CoV-2 is an uncommon infection in cats, amino acid sequences derived from SARS-CoV-2 proteins were mainly used as a model in this study [7]. As a result, they might also help to search for and predict other amino acid sequences with high affinity for FLAs. Although this is a preliminary exploration, we believe that these findings can be helpful in setting the basis for the characterization of the singular immune susceptibility of cats and for the in vivo screening of the most promising epitopes. Hence, these preliminary findings can be exploited as a tool for predicting CoVs’ sensibility in cats and, as a future perspective, for developing peptide vaccines able to stimulate the cell-mediated immune systems for untreatable diseases like FIP.

## Figures and Tables

**Figure 1 life-14-00334-f001:**
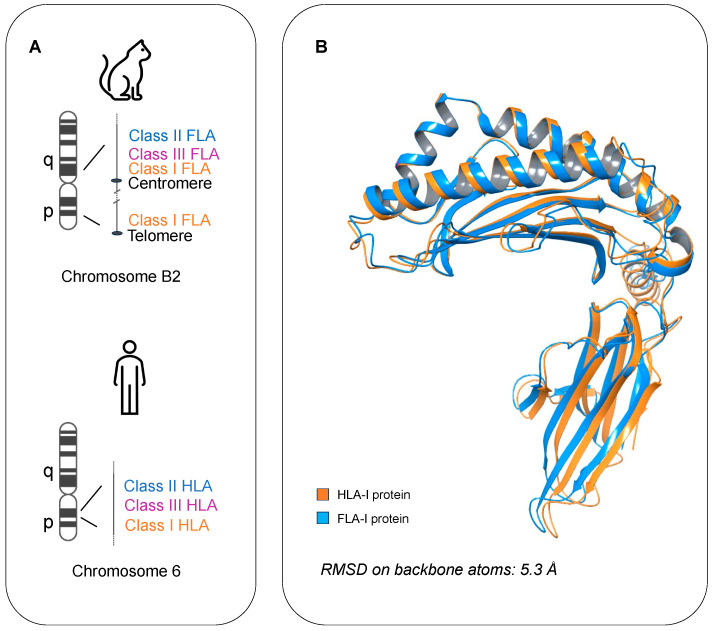
(**A**) Schematic representation of the MHC localization in the feline chromosome B2 (at the top) and the human chromosome 6 (at the bottom). (**B**) Superposition on the backbone atoms of two representative proteins encoded by class I HLA (orange ribbons) and FLA (blue ribbons).

**Figure 2 life-14-00334-f002:**
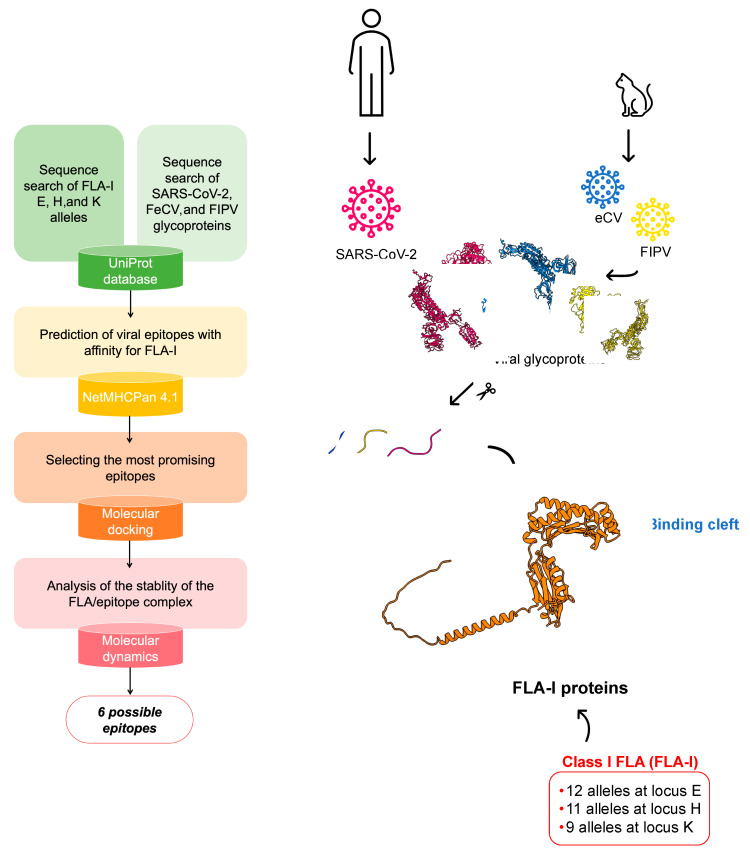
Workflow of the investigation.

**Figure 3 life-14-00334-f003:**
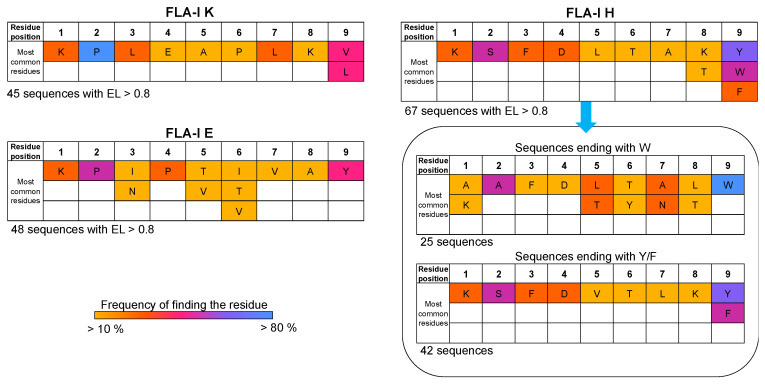
Sequence alignment of the peptides resulted in an EL score > 0.8 from NetMHCPan analysis. The frequency of finding the amino acid reported in the table in the corresponding position is indicated in a color scale (light orange for frequency > 10%, blue for frequency > 80%).

**Figure 4 life-14-00334-f004:**
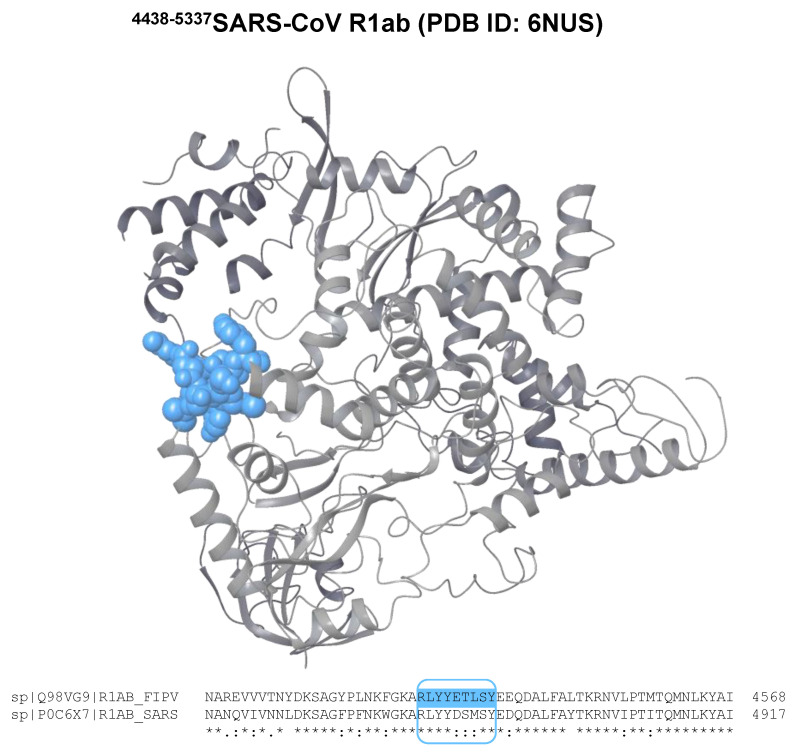
Cryo-EM structure of the moiety of SARS-CoV R1ab including the residues 4438–5337 (PDB ID: 6NUS [38]) in ribbon representation. The possible localization of _fipv-r_Ep4 is shown as blue CPK. The SARS-CoV R1ab sequence (UniProt ID: P0C6X7) aligned on FIPV R1ab (UniProt ID: Q98VG9, residues 4089–4988) reports 59.51% conserved residues in this domain and > 95% residues with similar chemical characteristics. The residues aligned reported at the bottom correspond to the moiety including _fipv-r_Ep4 (asterisk indicates conserved residues, colon indicates amino acids with high similarity, and dot indicates amino acids with low similarity).

**Figure 5 life-14-00334-f005:**
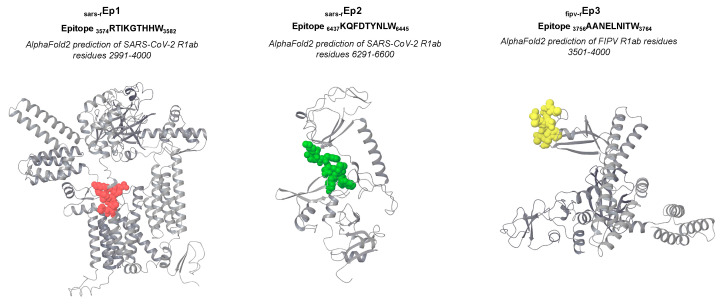
Analysis of the possible epitope exposition in SARS-CoV-2 R1ab and FIPV R1ab in moieties predicted with AlphaFold2. The possible localization of the epitopes is shown as colored CPK.

**Figure 6 life-14-00334-f006:**
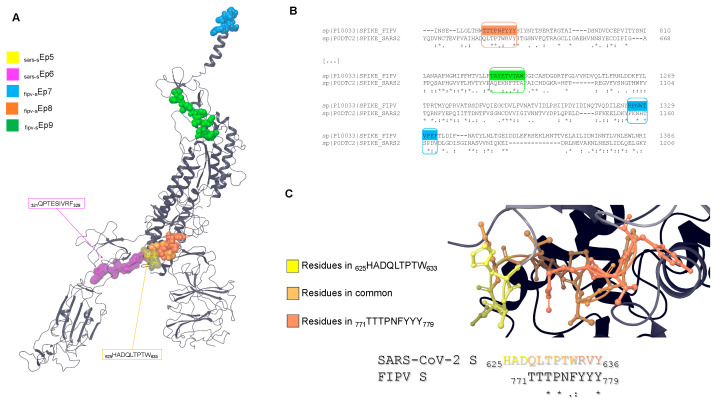
(**A**) Cryo-EM structure of the monomer of SARS-CoV-2 S (PDB ID: 7WEB [39]) in ribbon representation. The possible localization of the FIPV S-deriving epitopes is shown in CPK visualization, while the actual localization of SARS-CoV-2 S-deriving epitopes is shown as transparent surfaces. The SARS-CoV-2 S sequence (UniProt ID: P0DTC2, residues 612–1206) aligned on FIPV S (UniProt ID: P10033, residues 760–1386) reports 35.47% conserved residues in this domain and >70% residues with similar chemical characteristics. The residues aligned reported in (**B**) correspond to the moieties including _fipv-s_Ep7, _fipv-s_Ep8, and _fipv-s_Ep9 (asterisk indicates conserved residues, colon indicates amino acids with high similarity, and dot indicates amino acids with low similarity). (**C**) Focus on the overlapping residues in the possible location of _fipv-s_Ep8 with the real location of _sars-s_Ep5.

**Table 1 life-14-00334-t001:** Results of the molecular docking performed using ADCP on NetMHCPan–derived epitopes (EL > 0.8) in complex with NetMHCPan-derived FLA-I proteins. The peptides were sampled in extended and helix conformations. The results report the docking score energies (kcal/mol) and the number of poses generated by ADCP having an RMSD value < 6 Å.

				Extended Conformation		Helix Conformation	
Organism	Viral Glycoprotein	Epitope	FLA-I Receptor	Docking Score (kcal/mol)	Number of Poses with RMSD < 6 Å	Docking Score (kcal/mol)	Number of Poses with RMSD < 6 Å
SARS-CoV-2	R1a	RTIKGTHHW	FLA-I H*00501	−22.6	41	−22.5	40
SARS-CoV-2	R1ab	KQFDTYNLW	FLA-I K*00701	−20.8	27	−19.1	51
FIPV	R1ab	AANELNITW	FLA-I H*00501	−20.5	18	−19	31
FIPV	R1ab	RLYYETLSY	FLA-I H*00501	−19.8	22	−20.2	43
SARS-CoV-2	R1ab	KQFDTYNLW	FLA-I H*00501	−19.3	22	−17	32
SARS-CoV-2	R1a	RTIKGTHHW	FLA-I K*00701	−19.2	38	−18.7	35
SARS-CoV-2	R1a	RTIKGTHHW	FLA-I H*00401	−18.7	15	−19.6	7
FIPV	R1ab	YNLDIPHKL	FLA-I K*00701	−18.7	19	−18.3	26
SARS-CoV-2	R1a	RTIKGTHHW	FLA-I H*008012	−18.4	7	−18.1	8
SARS-CoV-2	R1a	VPFWITIAY	FLA-I E*00701	−17.1	6	−18.1	16
FIPV	R1ab	AANELNITW	FLA-I H*00401	−17	19	−18.1	25
SARS-CoV-2	R1a	VPFWITIAY	FLA-I K*00801	−16.8	20	−15.2	40
FIPV	R1ab	RLYYETLSY	FLA-I H*00401	−16.7	22	−18	23
FIPV	S	RPNWTVPEF	FLA-I E*00701	−16.6	21	−15.4	27
FIPV	S	TTTPNFYYY	FLA-I H*00501	−16.5	6	−16.3	44
SARS-CoV-2	R1a	VPMEKLKTL	FLA-I E*00701	−16.4	29	−15.8	22
SARS-CoV-2	R1a	RTIKVFTTV	FLA-I E*01101	−16.4	2	−14.7	21
SARS-CoV-2	S	HADQLTPTW	FLA-I E*01001	−16.3	55	−14.5	19
FIPV	S	RPNWTVPEF	FLA-I E*00101	−16.2	29	−13.2	21
SARS-CoV-2	R1a	VPFWITIAY	FLA-I E*00101	−16.1	30	−17.9	39
FIPV	R1ab	QNFDTYMLW	FLA-I H*00501	−16	12	−19.2	33
SARS-CoV-2	S	HADQLTPTW	FLA-I H*008012	−16	9	−14.5	3
SARS-CoV-2	S	HADQLTPTW	FLA-I E*00101	−15.9	17	−14.5	19
SARS-CoV-2	S	HADQLTPTW	FLA-I H*00401	−15.9	11	−16	23
SARS-CoV-2	S	QPTESIVRF	FLA-I E*00701	−15.6	14	−16.3	17
FIPV	S	TAYETVTAW	FLA-I H*00401	−15.5	7	−15.5	10
SARS-CoV-2	R1a	VPFWITIAY	FLA-I E*00501	−15.2	3	−15.2	7
SARS-CoV-2	R1a	VPMEKLKTL	FLA-I E*00101	−15.1	16	−14.5	20
SARS-CoV-2	R1a	VPFWITIAY	FLA-I H*008012	−14.8	16	−14.6	27
SARS-CoV-2	R1a	VPMEKLKTL	FLA-I K*00801	−14.8	31	−16.2	15
SARS-CoV-2	S	QPTESIVRF	FLA-I E*00501	−14.5	16	−14.5	2
SARS-CoV-2	R1a	VPMEKLKTL	FLA-I E*00501	−14.4	9	−14.2	23
SARS-CoV-2	S	QPTESIVRF	FLA-I E*00101	−14.4	32	−15.9	17
SARS-CoV-2	R1a	VPFWITIAY	FLA-I E*00501	−14.3	7	−14.2	24
SARS-CoV-2	R1a	LPSLATVAY	FLA-I E*00101	−14.1	20	−14.4	36
FIPV	R1ab	YPYGSGMVV	FLA-I K*00801	−13.8	35	−14.8	39
FIPV	S	RPNWTVPEF	FLA-I E*00501	−13.1	1	−11.7	5
FIPV	S	TAYETVTAW	FLA-I H*008012	−13.1	11	−13.6	38
FIPV	R1ab	RPIPDVPAY	FLA-I E*00501	−13	16	−11.3	26
SARS-CoV-2	S	QPTESIVRF	FLA-I E*00501	−12.7	1	−13.4	6
FIPV	R1ab	RPIPDVPAY	FLA-I E*00701	−12.5	11	−13.7	34
SARS-CoV-2	R1a	LPSLATVAY	FLA-I E*00501	−12.3	40	−13.2	7
FIPV	R1ab	RPIPDVPAY	FLA-I E*00101	−12.2	23	−10.5	20
FIPV	S	TAYETVTAW	FLA-I H*00601	−11.7	18	−13.7	20
SARS-CoV-2	R1a	VPMEKLKTL	FLA-I E*00501	−11.4	22	−13.2	12
FIPV	R1ab	RPIPDVPAY	FLA-I E*00501	−11.1	5	−10	15
SARS-CoV-2	R1a	LPSLATVAY	FLA-I E*00501	−10.1	11	−12.5	21

**Table 2 life-14-00334-t002:** Aliases for the best-ranked epitopes according to the NetMHCPan and molecular docking analyses.

**Epitope**	**Alias**
SARS-CoV-2 R1a _3574_RTIKGTHHW_3582_	_sars-r_Ep1
SARS-CoV-2 R1ab _6437_KQFDTYNLW_6445_	_sars-r_Ep2
FIPV R1ab _3756_AANELNITW_3764_	_fipv-r_Ep3
FIPV R1ab _4533_RLYYETLSY_4541_	_fipv-r_Ep4
SARS-CoV-2 S _625_HADQLTPTW_633_	_sars-s_Ep5
SARS-CoV-2 S _321_QPTESIVRF_329_	_sars-s_Ep6
FIPV S _1325_RPNWTVPEF_1333_	_fipv-s_Ep7
FIPV S _771_TTTPNFYYY_779_	_fipv-s_Ep8
FIPV S _1228_TAYETVTAW_1236_	_fipv-s_Ep9

**Table 3 life-14-00334-t003:** Results of the peptide search for epitopes deriving from FIPV glycoproteins in the UniProt database.

_fipv-r_Ep4 (FIPV R1ab _4533_RLYYETLSY_4541—_UniProt ID: Q98VG9)		
Retrieved from		
*Organism*	*Protein*	*UniProt ID*
Porcine transmissible gastroenteritis coronavirus (strain Purdue) (TGEV)	R1ab	P0C6Y5
Canine coronavirus	R1ab	A0A0D5ZXX1
Mink coronavirus strain WD1133	R1ab	D9J202
Swine enteric coronavirus	R1ab	A0A0U2LWJ9
Transmissible gastroenteritis virus	R1ab	C8YR34
**_fipv-s_Ep8 (FIPV S _771_TTTPNFYYY_779—_UniProt ID: P10033)**		
Retrieved from		
*Organism*	*Protein*	*UniProt ID*
Canine coronavirus (strain BGF10)	S	Q7T6T3
Canine coronavirus K378	S	Q65984
Canine coronavirus strain Insavc-1	S	P36300
Porcine transmissible gastroenteritis coronavirus (strain Miller) (TGEV)	S	P33470
Porcine transmissible gastroenteritis coronavirus (strain FS772/70) (TGEV)	S	P18450
Porcine transmissible gastroenteritis coronavirus (strain NEB72-rt) (TGEV)	S	Q01977
Porcine transmissible gastroenteritis coronavirus (strain Purdue) (TGEV)	S	P07946
**_fipv-s_Ep9 (FIPV S _771_TAYETVTAW_779—_UniProt ID: P10033)**		
Retrieved from		
*Organism*	*Protein*	*UniProt ID*
Canine coronavirus (strain BGF10)	S	Q7T6T3
Canine coronavirus K378	S	Q65984
Canine coronavirus strain Insavc-1	S	P36300
Porcine transmissible gastroenteritis coronavirus (strain Miller) (TGEV)	S	P33470
Porcine transmissible gastroenteritis coronavirus (strain FS772/70) (TGEV)	S	P18450
Porcine transmissible gastroenteritis coronavirus (strain NEB72-rt) (TGEV)	S	Q01977
Porcine transmissible gastroenteritis coronavirus (strain Purdue) (TGEV)	S	P07946
Mink coronavirus strain WD1133	S	D9J204
Porcine respiratory coronavirus (strain RM4)	S	P24413
Porcine respiratory coronavirus (86/137004/isolate British)	S	P27655

## Data Availability

The original contributions presented in the study are included in the article/Appendix A, further inquiries can be directed to the corresponding author/s.

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
