# Peer review of "An Exploratory Bioinformatic Investigation of Cats’ Susceptibility to Coronavirus-Deriving Epitopes"

_life, 2024, doi:10.3390/life14030334_

Round 1
Reviewer 1 Report
Comments and Suggestions for Authors
Buonocore et al. 2023 present a prediction of FLA-dependent susceptibility of felines to coronaviruses. The study is well explained and offers insight into possible requirements for coronavirus infection. Albeit, some open questions remain:
General
Figure 1 is present twice. This should be relabeled
Labeling of the figures is quite small which makes them difficult to read. This should be adjusted.
The author should consider a colorblind friendly color palette.
It could be interesting for the reader to see how similar FLA and HLA are – structurally and genetically.
For Epitope prediction – the authors only use NetMHCPan model. Can the authors also use other models to validate their findings? Moreover, it would be helpful for the reader to correlate the findings of the prediction with known HLA epitopes for cov2 from literature – this should be included in the discussion.
Figure 1
The authors show how the consensus residues for FLA-I look like. It would also be interesting to see how much the peptides with the best prediction listed on page 5 differ between the different viruses – are these homologous or highly conserved? How much variability do these epitopes have within the same virus species?
Molecular docking
Why did the authors use HPepDock and AutoDock CrankPep for molecular docking? Are there alternatives that could be used to validate these results?
Figure 2/3
In addition to projecting their epitope of choice onto the sars-cov2 structure, the authors should use alpha fold or ESM fold to model their protein of interest to then look for the localization of their epitope.
In the same line: How structurally similar are these proteins from the different viruses?
Given that some of these proteins come in different confirmations. Do the authors think that this may influence accessibility of the epitopes?
Molecular dynamics
Page 10 Line 287: Docking score should be reported
Discussion:
The authors should include a limitation statement and how their predictions could be validated with in vitro experiments.
The authors state: “Hence, these preliminary findings can be exploited as a tool for predicting CoVs’ sensibility in a wide number of species and, as a future perspective, for developing peptide vaccines able to stimulate the cell-mediated immune systems for un-treatable diseases like FIP.”
To support this hypothesis the authors should also include sequences from species that are not susceptible to sarscov2 infection.
Comments on the Quality of English Language
The writing quality is ok
Author Response
Buonocore et al. 2023 present a prediction of FLA-dependent susceptibility of felines to coronaviruses. The study is well explained and offers insight into possible requirements for coronavirus infection. Albeit, some open questions remain:
General
Figure 1 is present twice. This should be relabeled
We thank the reviewer for their availability to read and review our paper. Figure 1 and all the subsequent figures were relabeled and renamed in the main text accordingly.
Labeling of the figures is quite small which makes them difficult to read. This should be adjusted.
The author should consider a colorblind friendly color palette.
Figures 1,2,3 and 5 (now 2,3,4, and 6) were changed using colorblind-friendly palettes and with bigger labelings.
It could be interesting for the reader to see how similar FLA and HLA are – structurally and genetically.
We added a new figure (Figure 1) to compare the genomic organization and the most representative 3D structures of HLA and FLA.
For Epitope prediction – the authors only use NetMHCPan model. Can the authors also use other models to validate their findings? Moreover, it would be helpful for the reader to correlate the findings of the prediction with known HLA epitopes for cov2 from literature – this should be included in the discussion.
We added the validation of the epitopes via a search on the database IEDB in the discussion (lines 413-424)
Figure 1
The authors show how the consensus residues for FLA-I look like. It would also be interesting to see how much the peptides with the best prediction listed on page 5 differ between the different viruses – are these homologous or highly conserved? How much variability do these epitopes have within the same virus species?
We reported the analysis of the consensus in lines 217-225 by searching the obtained patterns in viral glycoproteins using Prosite.
Molecular docking
Why did the authors use HPepDock and AutoDock CrankPep for molecular docking? Are there alternatives that could be used to validate these results?
The docking simulation of peptides is often challenging using conventional programs like Autodock for docking small molecules. Indeed, the high number of rotatable bonds are not correctly sampled. Therefore, we used HPepDock and AutoDock CrankPep,validated in several papers for docking peptides. We added this explanation in lines 230-232.
Figure 2/3
In addition to projecting their epitope of choice onto the sars-cov2 structure, the authors should use alpha fold or ESM fold to model their protein of interest to then look for the localization of their epitope.
In the same line: How structurally similar are these proteins from the different viruses?
Given that some of these proteins come in different confirmations. Do the authors think that this may influence accessibility of the epitopes?
Modeling the protein with AlphaFold on ColabFolddis did not produce valuable results in a reasonable time range. Indeed, we preferred the epitope located on the experimental structure of the protein. The most accurate AlphaFold calculation resulted in pLDDT scores rather low ( Figure S1).
Molecular dynamics
Page 10 Line 287: Docking score should be reported
We added the docking scores in the indicated lines.
Discussion:
The authors should include a limitation statement and how their predictions could be validated with in vitro experiments.
We added this statement in lines 434-436.
The authors state: “Hence, these preliminary findings can be exploited as a tool for predicting CoVs’ sensibility in a wide number of species and, as a future perspective, for developing peptide vaccines able to stimulate the cell-mediated immune systems for un-treatable diseases like FIP.”
To support this hypothesis the authors should also include sequences from species that are not susceptible to sarscov2 infection.
We corrected the statement in lines 442-443. We greatly thank the reviewer for their scrupulous correction and the precious suggestions.
Reviewer 2 Report
Comments and Suggestions for Authors
In the manuscript” An exploratory bio-informatic investigation of cats’ susceptibility to coronavirus-deriving epitopes” (life-2868343) the authors performed an epitope mapping of nonapeptides deriving from SARS-CoV-2, FeCV and FIPV glycoproteins and predicted their affinities for different alleles included in the three main loci in class I FLAs (E, H and K).
The manuscript is interesting, however it should be improved.
I have a few minor concerns that can be easily addressed:
1) In the Table 1 is the docking score, what the value refers to? Is refers to best docked pose? There are many poses.
2) In the case of molecular dynamics simulation it is shown that for extended conformations the time of simulations is quite ok because the system is well equilibrated. However for the peptide sampled with helix conformation the authors should prolong the simulation because the system is not well equilibrated.
3) The discussion of the molecular dynamics should be therefore improved. What about energy from molecular dynamics simulations, any conclusions concerning stability of complexes?
4) There is double numbering in the reference list and empty reference number 43.
5) What is the table with epitopes under the Table 1 in the text? Is it separated from table 1?
Author Response
In the manuscript” An exploratory bio-informatic investigation of cats’ susceptibility to coronavirus-deriving epitopes” (life-2868343) the authors performed an epitope mapping of nonapeptides deriving from SARS-CoV-2, FeCV and FIPV glycoproteins and predicted their affinities for different alleles included in the three main loci in class I FLAs (E, H and K).
The manuscript is interesting, however it should be improved.
I have a few minor concerns that can be easily addressed:
- In the Table 1 is the docking score, what the value refers to? Is refers to best docked pose? There are many
We thank the reviewer for their comments. The table reports the poses of all the epitopes ranked according to their docking scores. All the docking scores lower than -10 kcal/mol are reported in the table. Therefore, sequences having more binding poses with a score lower than -10 are reported more than once. We better explained this in the main text (lines 235-236)
- In the case of molecular dynamics simulation it is shown that for extended conformations the time of simulations is quite ok because the system is well equilibrated. However for the peptide sampled with helix conformation the authors should prolong the simulation because the system is not well equilibrated.
In lines 333-335 we specified that MD simulations of the epitopes in helix conformations indicate low stability of the structures. Unfortunately, the limited HPC time machine hampered additional exploration of the dynamic behavior.
- The discussion of the molecular dynamics should be therefore improved. What about energy from molecular dynamics simulations, any conclusions concerning stability of complexes?
We added more details about the MD complexes in lines 403-406
- There is double numbering in the reference list and empty reference number 43.
Corrected.
5) What is the table with epitopes under the Table 1 in the text? Is it separated from table 1?
We added the caption for Table 2.
We thank the reviewer for their highly valuable comments.
Round 2
Reviewer 2 Report
Comments and Suggestions for Authors
The manuscript was corrected according to my suggestions therefore I recomend it for publication.